# A Mechanistic Study of Asymmetric Transfer Hydrogenation of Imines on a Chiral Phosphoric Acid Derived Indium Metal-Organic Framework

**DOI:** 10.3390/molecules27238244

**Published:** 2022-11-26

**Authors:** Xu Li, Ting Fan, Qingji Wang, Tongfei Shi

**Affiliations:** 1School of Light Chemical Engineering, Guangdong University of Technology, Guangzhou 510006, China; 2School of Chemistry and Chemical Engineering, South China University of Technology, Guangzhou 510641, China; 3College of Information and Communication Engineering, Hainan University, Haikou 570228, China

**Keywords:** asymmetric transfer hydrogenation, metal-organic framework, chiral phosphoric acid, density functional study, enantioselectivity

## Abstract

A density functional theory (DFT) study is reported to examine the asymmetric transfer hydrogenation (ATH) of imines catalyzed by an indium metal-organic framework (In-MOF) derived from a chiral phosphoric acid (CPA). It is revealed that the imine and reducing agent (i.e., thiazoline) are simultaneously adsorbed on the CPA through H-bonding to form an intermediate, subsequently, a proton is transferred from thiazoline to imine. The transition state **TS-*R*** and **TS-*S*** are stabilized on the CPA via H-bonding. Compared to the **TS-*S***, the **TS-*R*** has shorter H-bonding distances and longer C-H···π distances, it is more stable and experiences less steric hindrance. Consequently, the **TS-*R*** exhibits a lower activation barrier affording to the (***R***)-enantiomer within 68.1% *ee* in toluene. Imines with substituted groups such as −NO_2_, −F, and −OCH_3_ are used to investigate the substitution effects on the ATH. In the presence of an electron-withdrawing group like −NO_2_, the electrophilicity of imine is enhanced and the activation barrier is decreased. The non-covalent interactions and activation-strain model (ASM) analysis reveal that the structural distortions and the differential noncovalent interactions of TSs in a rigid In-MOF provide the inherent driving force for enantioselectivity. For −OCH_3_ substituted imine, the **TS-*S*** has the strongest steric hindrance, leading to the highest enantioselectivity. When the solvent is changed from toluene to dichloromethane, acetonitrile, and dimethylsulfoxide with increasing polarity, the activation energies of transition state increase whereas their difference decreases. This implies the reaction is slowed down and the enantioselectivity becomes lower in a solvent of smaller polarity. Among the four solvents, toluene turns out to be the best for the ATH. The calculated results in this study are in fairly good agreement with experimental observations. This study provides a mechanistic understanding of the reaction mechanism, as well as substitution and solvent effects on the activity and enantioselectivity of the ATH. The microscopic insights are useful for the development of new chiral MOFs toward important asymmetric reactions.

## 1. Introduction

Metal-organic frameworks (MOFs) have emerged as a special family of nanoporous materials. With enormous number of metal nodes and organic linkers available, the degrees of diversity and multiplicity in MOFs are far more extensive than any other porous materials. The crystalline structures, surface areas, pore sizes, and shapes in MOFs are readily tunable [1]. As a consequence, MOFs have attracted considerable interest for many potential applications. Particularly, there has been rapid development of MOFs as heterogeneous catalysts in the past several years [2,3]. Active catalytic sites can be introduced into MOFs for various reactions such as CO_2_ conversion, Michael addition, Knoevenagel condensation, etc. [4].

Recently, chiral MOFs have drawn increasing attention for asymmetric reactions [5]. A handful of chiral MOFs based on Lewis and Brönsted acid derivatives have been produced and tested. For example, Lin and coworkers systematically designed eight MOFs with chiral Lewis acidic sites and the MOFs were found to be highly active for asymmetric alkynylzinc additions and the enantioselectivities could be altered by tuning the channel size [6]. Using ligands derived from chiral phosphoric acid (CPA), they also reported a pair of highly porous chiral MOFs as active catalysts for Friedel-Crafts reactions between indole and imines [7]. Two chiral MOFs were prepared by Antilla and coworkers with CPA derived binol and ocMOM-1 in the channels and observed to possess higher enantioselectivity over parent ligands for transfer hydrogenation of benzoxazines [8]. Cui and coworkers designed 16 chiral MOFs of the same channel structures but different surface-isolated Lewis acid metal sites; these MOFs were proved to be a versatile family of heterogeneous catalysts for asymmetric allylboration, propargylation, Friedel-Crafts alkylation and sulfoxidation [9]. They further synthesized three indium-based MOFs with periodically aligned CPAs in the channels; the In-MFOs were found to possess significantly enhanced acidity over non-immobilized acids and exhibit high enantioselectivity for asymmetric condensation/amine addition and asymmetric transfer hydrogenation (ATH) of imines [10].

Most of the current studies on chiral MOFs, including the above-mentioned, are experimentally based, and there lacks microscopic understanding of reaction mechanisms. The intrinsic chiral environments in chiral MOFs provide confinement effects and specific interactions, leading to shape-, size-, chemo- and enantioselectivities that are not easily achievable in homogeneous catalysis. Thus, it is important to fundamentally and quantitatively unravel the roles of chiral environments in such MOFs for asymmetric reactions. At present, the first-principles based mechanistic studies in this field are scarce. Nevertheless, it should be noted that there have been numerous theoretical studies on homogeneous catalysis [11,12]. Goodman and coworkers applied density-functional theory (DFT) methods to examine the mechanism of Hantzsch ester hydrogenation of imines catalyzed by BINOL-CPAs, which were revealed to not only act as Brönsted acids to activate the imine groups but also interact with the Hantzsch ester leading to enantioselectivity [13]. Based on DFT calculations, they presented a model to describe the degree and sense of enantioselectivity of many reactions involving imines and BINOL-CPAs; the model rationalized the different factors (e.g., the (***E***)- or (***Z***)-preference of transition structures and the orientation of catalyst) governing enantioselectivity [14]. Furthermore, they proposed a computational approach to identify and quantify structural features for the effects of 3.3′-substituents on the enantioselectivity of BINOL-CPAs [15]. Shibata and Yamanaka reported a DFT study to examine ATH of ketimines and *α*-imino esters catalysed by BINOL-CPAs; the CPAs were revealed to activate the reactants to form cyclic transition structures and the high enantioselectivity was attributed to the steric interactions between the substituents in CPAs and substrates [16]. By both experimental and DFT techniques, Zhao and coworkers developed a highly selective ATH of *N*-aryl and *N*-alkyl ketimines using alcohol as hydrogen donor and a chiral iridium-CPA complex as catalyst [17].

In this study, a DFT study is reported to investigate the ATH of imines catalyzed by one of the In-MOFs (MOF 1) reported recently by Cui and coworkers [10]. The reaction is illustrated in Figure 1. We attempt to reveal the fundamental mechanism of ATH on the In-MOF, provide an in-depth understanding in the relationship of catalyst structure with catalytic activity and enantioselectivity. Meanwhile, the substitution and solvent effects are also considered. Following this introduction, the computational models and methods are described in Section 2. In Section 3, we first discuss the mechanism through different pathways and provide detailed analysis for the transitions states (TSs) in In-MOF; then, the substitution effects including electrophilic interaction and steric hindrance on the TSs are examined; the non-covalent interactions (NCI) and activation-strain model (ASM) analysis are used to reveal the original of enantioselectivity; finally, the solvent effects on the ATH is explored. In Section 4, the concluding remarks are summarized.

## 2. Results and Discussion

### 2.1. Mechanism

As mentioned above, we consider the ATH reactions for (***E***)-/(***Z***)-imine respectively with (***S***)-thiazoline as shown in Appendix A. Consequently, two TSs (**TS-*R*/-*S***) exist. Figure 1 illustrates the optimized TSs on the CPA site in rigid In-MOF. During the reaction, imine and thiazoline are simultaneously adsorbed on the CPA site through H-bonding to form co-adsorption complex as an intermediate; subsequently, thiazoline transfers its proton to the imine. The **TS-*R*** appears to possess a smaller molecular dimension than the **TS-*S***; consequently, the two TSs experience different steric effects. Appendix A lists the electronic energies and thermal corrections for the substituted imine, intermediates, TSs and products on the In-MOF in toluene. As shown in Figure 1a, there are two types of C-H···π interactions for the **TS-*R***: (1) edge-to-face C-H···π interaction between the C-H bond of phenyl ring in the TS and the π electron of aromatic ring on the In-MOF; (2) C-H···π interaction between the C-H bond of methyl group and the π electron of aromatic ring on the In-MOF. By contrast, there is only one type of C-H···π interaction for the **TS-*S*** as shown in Figure 1b. Table 1 lists the distances of C-H···π interactions (<3.5 Å) for the two TSs. Generally, the distances for the **TS-*S*** are shorter than those for the **TS-*R***, suggesting stronger steric hindrance for the **TS-*S***. Furthermore, Appendix A shows a few selected distances between the TSs and CPA. As listed in Appendix A, the **TS-*R*** has shorter H-bonding distances (O1-H1 and O2-H2) than the **TS-*S***. This implies the **TS-*R*** is more stable on the CPA site than the **TS-*S***. Indeed, one aromatic ring of imine in the **TS-*R*** is observed to point towards the cavity in the In-MOF.

Because of the above two factors, as listed in Table 2, the **TS-*R*** exhibits a lower activation barrier Δ*G*^‡^ (9.4 kcal/mol) than the **TS-*S*** (10.5 kcal/mol). Consequently, the (***R***)-enantiomer is preferentially produced in the ATH. The difference of activation barrier ΔΔ*G*^‡^ between the two TSs is 1.1 kcal/mol and the predicted *ee* is 68.1%. The experiment by Cui and coworkers observed that (***R***)-enantiomer was the major product with the *ee* of 99% (or 91% when Ketimines were in situ generated) [10]. It is worthwhile to note that the *ee* value is very difficult to be predicted accurately. With a small variation of 0.5 kcal/mol in ΔΔ*G*^‡^, the *ee* value would change by 36%. On this basis, the theoretical prediction may be considered being fairly good with the experiment.

### 2.2. Substitution Effects

The substitution in imine may cause two effects (electrophilic and steric) on the interaction and catalytic performance [18]. We consider three substituents −OCH_3_, −NO_2_, and −F. Appendix A lists the electronic energies and thermal corrections for the substituted imine, intermediates, TSs and products on the In-MOF in toluene. Figure 2 illustrates the frontier molecular orbitals of (***E***)-imines based on ωB97XD/6-31+G(d,p) method. For the lowest unoccupied molecular orbital (LUMO), the energy is lowest of −0.93 eV in −NO_2_ substituted imine as attributed to the strongest electrophilic effect by electron-withdrawing −NO_2_ group. In −F substituted, the LUMO energy is 0.07 eV. The substitution of electron-donating −OCH_3_ leads to the largest LUMO energy (0.17 eV) and HOMO energy (−7.52 eV), revealing that −OCH_3_ substituted imine has the weakest electrophilic and the strongest nucleophilic among the four. From Figure 2, the electrophilic effect is found to decrease in the order of −NO_2_ > −F > −H > −OCH_3_.

In general, a lower LUMO energy facilitates the ATH between imine (electrophile) and thiazoline (nucleophile). As listed in Table 2, the activation barriers Δ*G*^‡^ for −NO_2_ substituted imine are 6.7 and 9.7 kcal/mol for the **TS-*R/S***, respectively, the lowest among the four. The highest activation barriers Δ*G*^‡^ are 9.6 and 13.3 kcal/mol for −OCH_3_ substituted imine. Therefore, −NO_2_ substitution reduces the Δ*G*^‡^ for both TSs, enhances catalytic activity, and accelerates the ATH. On the contrary, the TSs with −OCH_3_ substitution have the lowest activity. These results are ascribed to the electrophilic property of substituent on the imine. When an electrophilic substituent (e.g., −NO_2_) is present in imine, the Δ*G*^‡^ is reduced and the reaction activity is improved. For the two TSs, as shown in Table 2 and Figure 3, the **TS-*R*** has a lower Δ*G*^‡^ than the **TS-*S*** for all the four imines. Thus, the formation of (***R***)-enantiomer is kinetically favored. Interestingly, the *ee* value is predicted to enhance from 68.1% to 98–99% when −H is substituted by −NO_2_, −F and −OCH_3_. Experimentally, high *ee* values (93~99%) were also observed for these substituted imines [10].

### 2.3. Original of Enantioselectivity

To quantitatively elucidate the enhanced enantioselectivity upon substitution, the stability and steric hindrance of TSs in rigid In-MOF, we adopted NCI analysis to visualize the non-covalent interactions as shown in Figure 4 and Appendix A. Since, ATH reaction has the highest 99.3% *ee* in −OCH_3_ substituted imine (as listed Table 2), we take TSs within −OCH_3_ substituted imine for example to illustrate the non-covalent interactions between TSs and rigid In-MOF (Figure 4). It shows that **TS-*S*** always has a large green surface than **TS-*R***, indicating that the high stabilizing π-stacking interactions are present in the benzene rings between (***E***)-imine and thiazole. However, **TS-*R*** within (***Z***)-conformation has a smaller steric hindrance to be the favoured TS, whose substrates are orientated away from the framework of In-MOF. Table 1 and Appendix A lists the C-H···π distances (<3.5 Å) and H-bonding and for the **TS-*R*** and **TS-*S*** of substituted imines. As shown in Figure 4 and Appendix A, the TSs of all the substituted imines are stabilized on the CPA site through strong H-bonding. The H-bonding distances in various substituted TSs exhibit similar values (Appendix A); however, the **TS-*R*** always has shorter H-bonding distances (O1-H1 and O2-H2) than the **TS-*S***, regardless of what the substituent is. On the other hand, the C-H···π distances in the **TS-*S*** are shorter than those in the **TS-*R*** for all the imines (Table 1). As the direct evidence shows that the **TS-*S*** with −OCH_3_ substituted imine has larger green surface (pink cycles in Figure 4) as compared with **TS-*R***, indicating that the **TS-*S*** suffers stronger steric hindrance from the rigid framework of In-MOF. Thus, the activation barriers Δ*G*^‡^ of the **TS-*S*** are higher than those of the **TS-*R*** for all the substituted imines. While the substitution is predicted to enhance the enantioselectivity, the degree of enhancement is different. As listed in Table 2, the ΔΔ*G*^‡^ and *ee* increase in the order of −H (1.1, 68.1% *ee*) < −F (3.0, 97.9% *ee*) ≈ −NO_2_ (3.0, 97.9% *ee*) < −OCH_3_ (3.7, 99.3% *ee*). Straightforwardly, the ΔΔ*G*^‡^ and *ee* for the formation of (***R***)-/(***S***)-enantiomers depend primarily on the steric hindrance of TSs from the rigid In-MOF, which can be described in a quantitative way through measuring the C-H···π distances. Intuitively, a shorter C-H···π distance leads to stronger steric hindrance. Among the four imines, the −OCH_3_ substituted one has the shortest C-H···π distances for the **TS-*S*** (Appendix A) and experiences the strongest steric hindrance as proved in Figure 4, the largest green surface (in pink cycle on left side). Consequently, the ΔΔ*G*^‡^ is the largest and the *ee* is the highest for −OCH_3_ substituted imine.

The relative energy differences (ΔΔE) between **TS-*R*** and **TS-*S*** are decomposed into contributions from the distortions of substrates in rigid In-MOF at the TSs geometries (ΔΔE_dist_) and the difference in non-covalent interactions between the substrates and In-MOFs (ΔΔE_int_), as listed in Table 3. For all of substituted imines, ***TS-R*** always has lower potential energy than **TS-*S***. In the original ASM model as reported by Bickelhaupt et al. [19,20], their catalysts are flexible and homogeneous chiral phosphoric acid. However, in this work, our catalyst is rigid In-MOF. We fix its atom in crystalline position when we optimized and searched the geometries of TSs. We consider that the rigid In-MOF can not distort its framework. As a result, ΔΔE_dist_ also can be decomposed into the energy difference required to distort substrates in different orientations (ΔΔE_dist-orien_) and conformations (ΔΔE_dist-conf_). As listed in Table 3, in the −OCH_3_ substituted imine, ΔΔE_dist-orien_ (5.6 kcal/mol) is the major contribution to ΔΔE. This indicates that the differential orientations of TSs in rigid In-MOF play a key role in enantioselectivity. Due to the large steric hindrance of −OCH_3_ group, TSs in different orientations are suffered different non-covalent interactions. However, in the −NO_2_ substituted imine, ΔΔE_dist-conf_ (−0.1 kcal/mol) and ΔΔE_dist-orien_ (−1.3 kcal/mol) are small, while ΔΔE_int_ is in a large value (5.7 kcal/mol). As shown in Appendix A, **TS-*R*** has a less green surface (blue cycle) with the longer C-H···π distances (as listed in Table 2) than **TS-*S*** (pink cycles). It reveals that the difference in non-covalent interactions between TSs and rigid In-MOFs is an important factor in controlling the enantioselectivity. In imine, ΔΔE_dist-orien_ (−4.7 kcal/mol) is a negative value, which compensates for the value of ΔΔE_dist-conf_. Finally, the values of ΔΔE_dist_ (−1.8 kcal/mol) and ΔΔE (1.1 kcal/mol) are small affording a low enantioselectivity. In −F substituted imine, ΔΔE_dist-orien_ (2.3 kcal/mol) corresponds closely to ΔΔE_dist-conf_ (3.6 kcal/mol) leading to a large value of ΔΔE_dist_ (5.9 kcal/mol). Obviously, the orientational and conformational distortion of TSs in rigid In-MOF for the enantioselectivity are crucial.

Based on the above results, we can conclude that the enantioselectivity of ATH reactions in In-MOF is mainly determined by the differential non-covalent interactions, the orientational and conformational distortion of TSs in rigid In-MOF. In summary, the structural distortions and the differential non-covalent interactions of TSs in a rigid framework provide the inherent driving force for enantioselectivity. This theoretical study reveals the relationship between catalyst structure and enantioselectivity, which is useful to facilitate the rational design and synthesis of chiral MOFs for the ATH and other asymmetric reactions.

### 2.4. Solvent Effects

All the above results are based on toluene as a solvent. To explore solvent effects, we further examine the ATH of −OCH_3_ substituted imine in dichloromethane, acetonitrile and dimethylsulfoxide. The dielectric constant *ε* rises from 2.37, 8.93, 35.69 to 46.83 in toluene, dichloromethane, acetonitrile and dimethylsulfoxide. Thus, toluene is the least polar, dichloromethane is intermediate, acetonitrile and dimethylsulfoxide are highly polar. Appendix A lists the electronic energies and thermal corrections for the intermediates, TSs and products in different solvents. As shown in Figure 5, the **TS-*R*** in any of the four solvents always has a lower activation barrier Δ*G*^‡^ than the **TS-*S***, leading to the favorable formation of (***R***)-enantiomer. With rising the dielectric constant *ε* of solvent, the Δ*G*^‡^ of each TS increases. Specifically, it increases from 9.6 (13.3) kcal/mol in toluene to 10.9 (14.2) kcal/mol in dimethylsulfoxide for the **TS*-R*** (**TS*-S***). Thus, the ATH is kinetically fastest in toluene. Furthermore, as seen in Figure 5, the ΔΔ*G*^‡^ decreases with rising *ε* and it has the largest value of 3.7 kcal/mol in toluene. Consequently, the enantioselectivity in toluene is the highest. From these results, the least polar toluene appears to be the best among the four solvents for the ATH. Such a phenomenon was experimentally observed with high conversion and enantioselectivity for ATH [21].

## 3. Computational Models and Methods

Figure 6a illustrates the crystalline structure of the In-MOF under this study. This MOF was prepared by Cui and coworkers from 3,3′,5,5′-tetracarboxylate ligands of chiral 1,1′-biphenol-2,2′-phosphoric acid [10]. The helical channels along the *a*-axis have a diameter of 1~1.2 nm, which would allow the diffusion of reactant and product into and out of the channels; moreover, the uncoordinated CPAs are periodically aligned within the channels as catalytic active sites. Therefore, the channels play an important role in catalytic performance and should be incorporated into the computational model. Because of the large structure and to reduce computational cost, a two-layer ONIOM (our own *n*-layered integrated molecular orbital and molecular mechanics) method [22] was adopted. As shown in Figure 6b, a cluster model with a channel was cut from the In-MOF and saturated by hydrogen atoms. The cluster had 776 atoms and consisted of 6 CPAs in the channel. Similar to our recent study [23], the CPAs including biphenyl rings and methyl groups were considered as the inner layer and treated quantum mechanically, while the remaining part of the cluster was the outer layer and described by a molecular mechanical approach. The reactant, intermediate, and TS and product were also in the inner layer (Figure 6c).

Imine may exist as either (***E***)- or (***Z***)-stereoisomer (Appendix A). The phenyl rings are located on the same side in (***Z***)-imine and on the opposite side in (***E***)-imine. The (***Z***)-imine has a smaller molecular dimension than the (***E***)-counterpart. In the ATH of imine with thiazoline on the In-MOF, we consider the reactions of (***E***)-/(***Z***)-imine with (***S***)-thiazoline, respectively. The inner layer including the cluster, reactant, intermediate, TSs and product was optimized by the commonly used exchange–correlation B3LYP functional [24] with 6-31G(d) basis set. The outer layer was fixed at its crystalline positions and mimicked by the universal force field (UFF) [25]. Energy minimum and TS were verified by frequency calculations also with B3LYP/6-31G(d) in the gas phase. Only one imaginary frequency was identified in each optimized TS (Appendix A). The thermal correction at 60 °C was obtained from the frequency calculations in the gas phase. The intrinsic reaction coordinate (IRC) [26] approach was adopted to confirm that the TS was connected to both reactant and product. In the experiment by Cui and coworkers, the ATH of imines was conducted in toluene [10]. Thus, electronic energy (*E*^ele^) was calculated with the polarizable continuum model [27] to mimic toluene. Based on the optimized geometries at ONIOM (B3LYP/6-31G(d) level: UFF) level, the single point energy (*E*^ele^) calculations were performed by using the polarizable continuum model within toluene. The Gibbs energy (*G*) and activation barrier (Δ*G*^‡^) were calculated from
*G* = *E*^ele^ + *G*^therm^(1)
Δ*G*^‡^ = *G*_TS_ − *G*_reactant_(2)
where *G*^therm^ is the thermal correction at 60 °C as in the experiment [10]; *G*_TS_ and *G*_reactant_ are the Gibbs energies of TS and reactant, respectively.

Various density functionals have been used to explore catalytic mechanism in porous materials such as zeolites and MOFs [23,28,29,30]. To examine the effects of DFT method, *E*^ele^ was calculated by four different functionals (M06-2X [31], M06-L [32], ωB97XD [33] and B3LYP-D3 [34]) at 6-31+G(d,p) basis set in toluene. As shown in Equations (1) and (2), the total energy was the sum of the thermal correction *G*^therm^ in the gas phase at ONIOM (B3LYP/6-31G(d) level: UFF) level, and the single point energies (*E*^ele^) within toluene based on different functionals at 6-31+G(d,p) basis set. Appendix A lists the *E*^ele^ and *G*^therm^ for the cluster, reactant, co-adsorption complex (i.e., intermediate), TS, and product in toluene. Appendix A shows the relative Gibbs energies via two **TS*-R*** and **TS*-S***, with the isolated reactant plus the cluster as a reference. Among the four functionals, ωB97XD accounts for proper electron correlation and predicts the relative Gibbs energies ranging from −25 to −60 kcal/mol. M06-2X yields prediction close to ωB97XD, whereas B3LYP-D3 and M06-L exhibit large differences from ωB97XD and M06-2X. Especially, the relative energies of species obtained by B3LYP-D3 functional were in a range from −132.1 to −163.4 kcal/mol (in Appendix A), these so huge energies were impossible in the experiment. This result means that B3LYP-D3 functional can not describe correctly In-MOF structure due to the excessive dispersion correction for the long-range interactions. While the relative Gibbs energies are sensitive to the functional, the activation barriers are not (except for M06L). As illustrated in Figure 7, the activation barriers from M06L are 1.9 and 4.5 kcal/mol via **TS*-R*** and **TS*-S***, respectively; the other three functionals (M06-2X, B3LYP-D3 and ωB97XD) predict the activation barriers between 7.2 and 10.5 kcal/mol. The activation barrier based on the M06-L functional was too small with the value of 1.9 kcal/mol, which means this reaction would be very fast. However, this phenomenon was not mentioned by Cui and coworkers [10]. Nevertheless, all the four functionals reveal that the pathway via **TS*-R*** has a lower barrier than **TS*-S*** and hence it is kinetically more favorable. In addition, the energies obtained from M06-2X and ωB97XD functionals were reasonable. ωB97XD provides the middle values of energies as compared with the other functionals, thus, we finally used ωB97XD was used for all the calculations, unless otherwise stated. This functional was demonstrated to be well suited for the description of non-covalent interactions for zeolites and MOFs [30,35,36].

To elucidate the polarity effect, three other solvents were also considered including dichloromethane, acetonitrile and dimethylsulfoxide with *ε* of 8.93, 35.69 and 46.83, respectively. Moreover, we examined the substitution effects by varying the group R in imine from –H to −OCH_3_, −F and −NO_2_. Among the three substituents, −OCH_3_ is electron-donating, whereas −F and −NO_2_ are electron-withdrawing, and their effects on catalytic performance were quantitatively evaluated. All the DFT calculations were carried out using Gaussian 09 [37].

For the ATH of imine to form (***R***)-/(***S***)-amines, the macroscopic rate constant *k_R_*_/*S*_ can be derived from the transition-state theory [38]
(3)$$k_{R/S}=\frac{k_{B}T}{h}e^{-\Delta G_{R/S}^{‡}/\mathrm{RT}}$$
where *k*_B_ is the Boltzmann constant, *h* is the Plank’s constant, *T* is temperature, and $G_{R/S}^{‡}$ is the activation barrier. The enantioselectivity of (***R***)-/(***S***)-enantiomers is quantified by enantiomeric excess (*ee*%)
(4)$$ee\%=\frac{\left[k_{S}\right]-\left[k_{R}\right]}{\left[k_{S}\right]+\left[k_{R}\right]}\times 100\%=\frac{1-e^{-\Delta \Delta G^{‡}/RT}}{1+e^{-\Delta \Delta G^{‡}/RT}}\times 100\%$$
where $\Delta \Delta G^{‡}=\left|\Delta G_{S}^{‡}-\Delta G_{R}^{‡}\right|$ is the difference between the activation barriers for the formation of (***R***)-/(***S***)-enantiomers.

The Non-covalent interactions (NCI) [39] index method was used to reveal the isosurface of non-covalent interactions. The reduced density gradient (RDG) was obtained by Multiwfn [40]. The RDG function was expressed in Equation (5), where *ρ*(r) was the total electron density. Due to a large number of atoms in In-MOF structure, the single point energy calculations of TSs in In-MOF were unable to load and achieve the total electron density. The cluster model was cut from the optimized geometry (as shown in Figure 1c) at ONIOM (B3LYP/6-31G(d) level: UFF) level. The single point energy calculation based on this cluster model was carried out at ωB97XD/6-31+g (d,p) level. Different types of interactions (attractive and repulsive) were distinguished by multiplying the density with the sign of the second-density Hessian eigenvalue (λ2). The sign of λ2 distinguishes the bonded (λ2 < 0) from nonbonded (λ2 > 0) interactions. The isosurface of RDG was plotted by using VMD software [41].
(5)$$\mathrm{RDG}(r)=\frac{1}{2{\left(3\pi^{2}\right)}^{1/3}}\;\frac{\left|\nabla \rho \left(r\right)\right|}{\rho {\left(r\right)}^{4/3}}$$

The origin of enantioselectivity was analyzed by using the activation-strain model (ASM), as reported by Bickelhaupt et al. [19,20] (or, the distortion-interaction model of Houk and Ess [42,43]). We executed the single point energy calculation based on the cluster model at ωB97XD/6-31+g(d,p) level to obtain the potential energy. The potential energy surface was calculated from:(6)$$\Delta E\left(\zeta \right)=\Delta E_{\mathrm{dist}}\left(\zeta \right)+\Delta E_{\mathrm{int}}\left(\zeta \right)$$
where ζ is the reaction coordinate, ΔE_dist_ (ζ) is the distortion energy, which was associated with the structural deformation that the substrates undergo, ΔE_int_ (ζ) was the interaction between these increasingly deformed substrates. The activation energy of a reaction $\Delta E^{‡}=\Delta E\left(\zeta^{\mathrm{TS}}\right)$ consists of activation strain $\Delta E^{‡}{}_{\mathrm{dist}}=\Delta E_{\mathrm{dist}}\left(\zeta^{\mathrm{TS}}\right)$ plus the TS interaction $\Delta E^{‡}{}_{\mathrm{int}}=\Delta E_{\mathrm{int}}\left(\zeta^{\mathrm{TS}}\right)$:(7)$$\Delta E^{‡}=\Delta E^{‡}{}_{\mathrm{dist}}+\Delta E^{‡}{}_{\mathrm{int}}$$

The distortion energy was the sum of two components: the energy required to distort substrate from ground-state geometry to transition-state geometry and the energy required to distort catalyst from ground-state to transition-state geometry:(8)$$\Delta E_{\mathrm{dist}}=\Delta E_{\mathrm{dist}-\mathrm{sub}}+\Delta E_{\mathrm{dist}-\mathrm{cata}}$$

## 4. Conclusions

We have conducted DFT calculations to investigate the ATH of imines catalyzed by a chiral In-MOF. While the relative Gibbs energies are found to strongly depend on the functional used, the activation energies remain nearly same. The optimized **TS-*R*/*S*** are stabilized on the CPA site via H-bonding. Compared to the **TS-*S***, the **TS-*R*** has shorter H-bonding distances. On the other hand, two types of C-H···π interactions exist for the **TS-*R***, however, there is only C-H···π interaction for the **TS-*S***. The distances of C-H···π interactions for the **TS-*R*** are longer than those for the **TS-*R***. Overall, the **TS-*R*** is more stable and experiences less steric hindrance than the **TS-*S***. As a result, the activation barrier via the **TS-*R*** is 1.1 kcal/mol lower than that via the **TS-*S*** in toluene, and (***R***)-enantiomer is preferentially formed within 68.1% *ee*. By substituting –H in the imine by electron-withdrawing group like −NO_2_, the activation barrier is reduced due to stronger electrophilicity. However, the enantioselectivity also depends on H-bonding and steric hindrance between the TS and In-MOF. The non-covalent interactions and activation-strain model analysis reveal that the structural distortions and the differential non-covalent interactions of TSs in rigid In-MOF provide the inherent driving force for enantioselectivity. Among the substituted imines considered, the **TS-*S*** with −OCH_3_ substituent has the shortest C-H···π distance and hence the strongest steric hindrance; consequently, the difference in the activation barriers becomes the largest, leading to the highest enantioselectivity. The predicted *ee* values are 98–99% for −NO_2_, −F, and −OCH_3_ substituted imines. Furthermore, it is found that a less polar solvent is beneficial to the ATH with faster kinetics and higher enantioselectivity. Among the four solvents (toluene, dichloromethane, acetonitrile and dimethylsulfoxide) under study, toluene appears to be the best. The theoretical predictions match fairly well with experimental results. This study clearly reveals the mechanism of ATH in the In-MOF, demonstrates the relationship between catalyst structure and performance, and it may assist in the rational design of new chiral MOFs for high-performance catalysis.

## Data Availability

Not applicable.

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
