# Peer review of "A Mechanistic Study of Asymmetric Transfer Hydrogenation of Imines on a Chiral Phosphoric Acid Derived Indium Metal-Organic Framework"

_molecules, 2022, doi:10.3390/molecules27238244_

Round 1
Reviewer 1 Report
In the manuscript by Li, Fan, Wang and Shi a theoretical study on the asymmetric transfer hydrogenation of imines catalyzed by an indium metal-organic framework is reported.
The study revealed that the imine and the reducing agent thiazoline are adsorbed on the catalyst through H-bonding to form an intermediate that subsequently experiences a H transfer from the reducing agent to the imine. Two possible mechanisms, via the so-called TS-R and TS-S, are studied, and it is concluded that the mechanism through TS-R is kinetically favored, in agreement with the experimentally observed enantiomeric excess. The authors also include a comparison between differently substituted imines and between different solvents, and the conclusions reached are in fairly good agreement with the experimental data.
The manuscript describes an interesting work and deserves publication, although some minor points and clarifications should be provided by the authors.
- The authors explain that they use a two-layer ONIOM method. For the reactants (cluster and thiazoline), intermediate, transition state and product, the inner layer has been optimized at B3LYP/6-31G(d) level, and further frequency calculations and IRC have been conducted at the same level. However, it is not clear if the authors conduct these optimizations and frequency calculations in toluene (with the SCRF=(PCM, solvent=toluene) option in the gaussian command line), or if they made the optimizations and frequency calculations in vacuo, and after that they conduct single-point calculations including the solvent, on the previously obtained geometries. The authors should clarify this point.
- By reading the last three lines on page 3, and equations (1) and (2), it seems that the authors obtain the Gibbs energies by summing the electronic energy, obtained from single-point calculations that include the solvent, to the G thermal correction at 60 degrees, obtained from the in vacuo optimizations. But this is not explicitly explained by the authors, if this is the case. In any case, the authors should better clarify how they obtain their G values.
- On page 4, the authors say that they use up to four different functionals at 6-31+G(d,p) in toluene to obtain energetic values. Again, it is not clear how exactly the authors do this. Looking at Table S2 it seems that the authors take the thermal correction to G from their B3LYP/6-31G(d) calculations (in gas-phase?) and the electronic energies from single-point calculations, that include the toluene, on the geometries previously obtained in vacuo at the B3LYP/6-31G(d) level. But this is not explicitly stated by the authors. If this was the case, the electronic energies at the four different functionals (FUNC) used have been obtained at the FUNC(toluene)/6-31+G(d,p)//B3LYP(gas-phase)/6-31G(d), while the thermal correction to G was obtained at B3LYP(gas-phase)/6-31G(d). Is this the case? The authors should further clarify these points.
- In Figure 2, I see that the right profile corresponds to TS-R, while the left one to (a)(b) TS-S. This is probably due to wrong pdf-conversion, or wrong pdf reading in my computer, because it is the other way around.
- The authors conclude that the wB97XD functional is the best suited for their calculations. It is not really clear why. The authors explain that this method “predicts the relative Gibbs energies ranging from -25 to -60 kcal/mol”. Are these values in agreement with experimental data? If this is the case, it should be stated. If not, the authors should better explain why they choose to continue the calculations with this functional.
- In Figure 3 the TS-R and TS-S are depicted. It is very difficult to see where the H being transferred is in the ball&stick representations. Could the authors depict this H atom with a different color? This also happens in Figure 6.
- The enantiomeric excess obtained by the authors in page 6 is 68.1%. This value is not very close to the experiment, which is 99%, or 91% when Ketimines were in situ generated. However, if the authors would take their B3LYP-D3 values in Table S2, an enantiomeric excess of ca 89.6% would be obtained. And if they would take their M06-L values, the obtained enantiomeric excess would be of 96.4%. Hence, maybe the authors could have used their M06-L values, or even their B3LYP-D3 values, because they better approach the experimental enantiomeric excess, instead of their wB97XD values. Have the authors considered these possibilities?
- In Figure S2 four profiles appear with different values for the M06-L lines. Probably only the two on the upper part should be there.
Reviewer 2 Report
With the surname Brönsted, there are always some peculiarities in spelling. Please choose one way. On the second page of the manuscript, there are different ways of writing.
Why was the calculation of the product in toluene chosen? Although other solvents are also explored in the manuscript. This should be clarified.
Reviewer 3 Report
(1) The authors clearly no aim of this study except few things like solvent effect and effect of different groups on stereoselectivity calculated using already published work (Angew. Chem. Int. Ed. 2019, 58, 14748). In this published paper, authors have already described both heterogeneous and homogenous (model systems) and full mechanisms.
(2) My suggestion is, you can publish this work, after finding the origin of stereoselectivity using AIM analysis (non-covalent interactions observed through BCPs), and Strain analysis (distortion and interaction energies variation among given Substituents). Then this aim of the paper should be meaningful and maybe suitable to publish.
(3) HOMO-LUMO only not sufficient to explain stereoselectivity, there other factors also needs to be considered like strain and all.
Round 2
Reviewer 3 Report
Authors have changed, major changes and explained stereoselectivity based on the Distortion-Interaction model. But still small changes required as shown below.
Major changes:
(1) In Table 2: Mentioned both Experimental and DFT respective ee's. If experimental proved with -H (already Exp value is available and DFT also proved: cite Angew. Chem original paper), and maybe -F, -NO2 and -OCH3 only show DFT ee values (if no exp values, just mentioned).
Minor Changes:
(1) In Figure 6 (Non-Covalent Interactions) High ee (-OCH3 = 99.3) combination only should show in the main Manuscript for clear picture and remaining (-H, -F and -NO2) can show in Supporting Information.
(2) In Table 3; Distortion energies should be in Positive sign and Interaction Energies should be in Negative Sign (even though del of Del E). Just cross check.
(3) If Table 3, relative energies are correct, then mentioned as Note, with respect to lowest stereoisomer for example "R".
